# Setting the Tibial Component Rotation Based on Femoral Landmarks Allows Congruent Knee Kinematics in Robotic-Assisted Medial Unicompartmental Knee Replacement

**DOI:** 10.3390/jpm13040632

**Published:** 2023-04-04

**Authors:** Matteo Innocenti, Luigi Zanna, Mustafa Akkaya, Kim Huber, Bernhard Christen, Tilman Calliess

**Affiliations:** 1Department of Orthopaedics, University Hospital of Florence, 50139 Florence, Italy; matteo.innocenti@unifi.it (M.I.);; 2Ankara City Hospital, 06800 Ankara, Turkey; 3Articon Spezialpraxis für Gelenkchirurgie, 3013 Berne, Switzerland

**Keywords:** unicompartmental knee arthroplasty, rotation, tibial component, axial alignment, CT scan, robotic

## Abstract

The accurate positioning of the prosthetic components is essential for achieving successful results in medial unicompartmental knee arthroplasty (mUKA). The tibial component rotation in image-based robotic-assisted UKA is usually based on tibial bony landmarks matched to the pre-operative CT model. The study aimed to evaluate whether setting the tibial rotation on femoral CT-based landmarks allows congruent knee kinematics. We retrospectively analyzed data from 210 consecutive image-based robotic-assisted mUKA cases. In every case, we set the tibia rotation landmark parallel to the posterior condylar axis and centered it on the trochlea groove defined on the preoperative CT scan. The implant positioning was primarily set parallel to this rotation landmark and then adjusted based on tibial sizes avoiding component over- or under-hang. During surgery, we recorded the knee kinematics under valgus stress to reduce the arthritic deformity. A femoral-tibial contact point was recorded over the entire range of motion and displayed as a tracking profile on the tibia implant. The femoro-tibial tracking angle (FTTA) was then calculated based on a tangent line to the femoro-tibial tracking-points and the difference to the femur-based rotation landmark. In 48% of the cases, we could position the tibia component exactly to the femoral rotation landmark, whereas in 52% of cases, minimal adjustments were made to avoid component’s under- or over-hang. The mean tibia component rotation (TRA) with reference to our femur-based landmark was +0.24° (SD ± 2.9°). The femur-based tibia rotation landmark showed a high correspondence to the FTTA with 60% of the cases having less than 1° of deviation. Mean FTTA was +0.7° (SD ± 2.2°). The mean difference between the absolute value of the TRA and the FTTA (|TRA| − |FTTA|) was −0.18° (SD ± 2°). Setting the tibial component rotation based on CT scan femoral landmarks and not on tibial anatomical landmarks is a reliable method to obtain congruent knee kinematics during image-based robotic-assisted medial UKA with less the 2° deviations on average.

## 1. Introduction

Medial unicompartmental knee arthroplasty (mUKA) has been shown to be a successful procedure for single-compartment osteoarthritis treatment, offering functional advantages over total knee arthroplasty, such as bone preservation, less morbidity, quicker return to activity, and more natural prosthetic knee kinematics [1,2,3,4]. However, mUKA is technically demanding, and a correct surgical procedure along with an optimal component positioning are essential to obtain satisfactory outcomes and to prevent early mUKA failures [5,6]. The implant positioning is the key to achieve good clinical and functional outcomes, as this determines directly the knee kinematics – much more than the implant design does. Components’ malpositioning is often correlated with suboptimal femoro-tibial contact points. This can potentially increase the intercompartmental stresses that may accelerate polyethylene wear and increase the revision rate, as described earlier [7,8,9]. 

Although the influence of components’ positioning in the three special planes has been widely described in radiological research on total knee arthroplasty (TKA), the literature with regard to evaluating the impact of tibial component alignment in every single plane for mUKA is still very limited. The majority of these studies have only analyzed the effects of tibial component positioning in the sagittal and coronal plane focusing on the varus/valgus angulation, the joint line height or the tibia slope as factors influencing knee kinematics and implant survival. However, less is known about the correct rotation positioning of the tibia component and its potential impact in terms of mUKA failures, stiffness, the reduction in patient reported outcome measures (PROMs), and unexplained pain [4,5]. Moreover, there is currently an absence in the literature concerning the relationship between the combined femoro-tibial rotation alignment and the resulting knee kinematic during the entire range of motion (ROM). 

During surgery as well as on standard postoperative radiographically imaging, the tibia component rotation or the combined femoro-tibial alignment can not be precisely determined. Also, in traditional image-less navigation the axial alignment relies on more or less subjective parameters defined by the surgeon. This could be addressed by the use of image-based robotic systems that are meant to provided three-dimensional image data to improve the surgeon’s ability to objectively define the surgical landmarks and thus to reproduce alignment, accurate rotational and sagittal components positioning, and congruent femoro-tibial kinematics during the ROM. This has the potential to decrease the early failure rate due to implant mispositioning [10,11]. Nevertheless, compared with the coronal and sagittal alignment of the knee, the correct rotational position is still difficult to achieve, particularly in small operating fields with mini-incision surgery, which is usually the surgical approach used for mUKA as part of respecting the minimally invasive surgery principle [12]. Accurate femoro-tibial and rotational alignment are difficult to obtain based on objective bony landmarks [13,14,15], even if supported by the robotic-assisted arm and matched to the pre-operative CT scan model. Especially, setting the tibia component rotation based on tibia landmarks appears error prone due to the large heterogeneity of the proximal tibia anatomy. Taking into account that the positional relationship between the femoral and tibial components would be the most important factor for rotational alignment and femoro-tibial tracking during the whole range of motion (ROM) in mUKA [12], the purpose of this study was to evaluate whether copying the rotational positioning of the tibial component with the femur by setting the tibial rotation on femoral CT-based landmarks (femoral-referenced landmarks) instead of tibial bony landmarks allows congruent knee kinematics. 

Our hypothesis is that setting the tibial component rotation based on CT scan femoral landmarks and not on tibial anatomical landmarks is a reliable method to obtain congruent knee kinematics during image-based robotic-assisted medial UKA.

## 2. Materials and Methods

Intra-operative image data elaborated by the MAKOplasty software of 210 consecutive patients undergoing imaged-based robotic-assisted mUKA from January 2019 to September 2022 at a single institution were retrospectively reviewed. The indications for a mUKA were the presence of an isolated medial compartment osteoarthrosis with a Kellgren–Lawrence [16] grade III or IV score with a reducible deformity on the coronal plane, the absence of antero-posterior or medio-lateral knee instability, a full range of motion or a range of motion with maximum 10° of flexion contracture, and the absence of symptoms on the other knee’s compartments. Patients who underwent any other simultaneous associated surgical procedures were excluded. 

### 2.1. Preoperative Planning

A Preoperative CT-scan of the hip, knee, and ankle was conducted following the standard MAKOplasty protocol and uploaded to STRYKER proprietary platform (Stryker, Mahwah, NJ, USA). Based on this scan, an individual knee model was segmented by the on-site MAKO product specialist (MPS) and used for the primary surgical planning according to the STRYKER standard protocol. Only the tibia-rotation landmark was not set based on tibia landmarks, as suggested, but on the femoral anatomy. In the menu to set the tibia rotation, the scan was scrolled up to the femur and the landmark was then co-aligned to the posterior femur condyles centered in the middle of the notch (Figure 1). Here, the anterior arrow should match the trochlea groove. If there was a relative internal rotation to the trochlea groove, the rotation was adjusted until it met the groove and was captured as the “new” femur-based tibia rotation landmark (Figure 1).

In the definite implant planning, the tibia component was set to 0° compared to this femur-based rotation landmark (Figure 2). Then, tibia component size was determined and the tibia rotation was fine-tuned to avoid component over- or under-hang (Figure 3a,b). The other alignment parameters were set to reconstruct the natural slope up to 7° maximum and the joint-line obliquity up to 3° varus. The proudness of the implant was pre-set at 4 mm.

The femoral implant was planned according to the standard protocol to best reconstruct the natural surface anatomy with a proudness between 2.0 and 2.5 mm to start with.

### 2.2. Surgical Technique

Two knee surgeons highly trained in image-based robotic-assisted surgery performed all procedures. A mini medial-parapatellar approach was used for every case, and the same fixed-bearing metal-backed cemented unicompartmental knee prosthesis was implanted (RESTORIS MCK partial knee, Stryker, Mahwah, NJ, USA) with the assistance of the MAKO robotic system (Stryker, Mahwah, NJ, USA). An examination of all knee compartments and the integrity of the anterior cruciate ligament was performed to confirm the indication for mUKA and proceed with the surgery. A classical workflow for a MAKO robotic-assisted surgery was used to position the arrays and to register and match the bone anatomy. 

After that, all osteophytes were resected and the range of motion (ROM) and the kinematic data of the knee were acquired by applying a varus–valgus stress to reduce the arthritic deformity. The reciprocal femoro-tibial contact points (femoro-tibial tracking) were recorded and displayed during the entire ROM. This tracking profile was recorded as screen shots for further analysis.

### 2.3. Data Acquisition

The MAKOplasty software and the Aequo software version 1.9.1 were used to collect and elaborate the intra-operative data. Two independent observers made all the measurements and calculations. 

First, the degree of the tibial component rotation in the pre-planning was recorded in relation to femur-based tibia rotation landmark (tibia rotation angle, TRA) as shown in the screenshots (Figure 2, Figure 3 and Figure 4). Positive values describe external rotation while negative values describe internal rotation in the axial plane. We recorded the number of cases in that the tibial component position in the axial plane had to be fine-tuned/rotated by more than +1° or less than −1° in order to avoid component over- or under-hang. 

Second, the femoro-tibial tracking angle (FTTA) in relation to the femur-based tibia rotation landmark was calculated (Figure 4). A line passing across at least 4 consecutive femoro-tibial tracking points displayed on top of the tibial component surface on a monitor screenshot taken in between 0° and 10° of knee extension was drawn. A second line tangent and parallel to the lateral edge of the tibial component was added on the same monitor screenshot. The angle between those two lines was then calculated and subtracted from the TRA to get the FTTA with respect to the landmark (Figure 4). 

### 2.4. Statistical Analyses

The statistical analysis was performed using SPSS^®^ statistics software (IBM^®^, Armonk, New York, NY, USA). The two-sided Grabb’s test was used to detect the presence of outliers in the TRA and FTTA considering a Z value > 3.5 as significant to define outliers. The mean, median, and standard deviations were calculated for the TRA and the FTTA. The Student’s *t*-test was used to compare the difference between TRA and FTTA taking a *p* value < 0.05 as statistically significant. 

The intra- and inter-observer reliability for FTTA was assessed using intraclass correlation coefficients (ICC) with a two-way mixed effect model. 

## 3. Results

In 48% of the cases, we could position the tibia component exactly to the femoral-based tibia rotation landmark, while the other 52% of cases required adjustments for the tibia component rotation of more than +1° (mean + 3.2°, SD ± 2.2°) or less than −1° (mean − 2.9°, SD ± 1.9) in the axial plane to match the tibial anatomy, avoiding components’ over- or under-hang. The mean TRA was +0.24° (median + 0.2°, SD ± 2.9°) with respect to the rotation landmark. One outlier was detected for the TRA and it was a case with a TRA of +14° (Z value 4.63). All other cases were within a ±5° range.

The mean FTTA was +0.7° (median + 0.2°; SD ± 2.2°) (Table 1). A total of 57% of the cases had less than 1° of deviation from the rotation landmark. A single outlier was detected for the FTTA and it was at −11.2° (Z value 4.13). All other cases were within a ±5° range.

The mean difference between the absolute value of the TRA and the FTTA (|TRA| − |FTTA|) was −0.18° (median 0°, SD ± 2°). No significant difference was found between the TRA and the FTTA (*p* = 0.069).

The ICC of FTTA showed excellent agreement between raters. Interrater reliability of the degrees of FTTA was 0.93 (95% CI: 0.89–0.96).

## 4. Discussion

In the present study, we showed that an optimal congruent femoro-tibial knee kinematic in the robotic-assisted mUKA could be restored by positioning the tibial component in the axial plane based on CT-scan femoral anatomical landmarks independently from pre- and intra-operative tibial anatomical landmarks. We only found a single outlier for both the TRA and the FTTA, with an average value of those two angles of about 0° (+0.24° and +0.7°, respectively) in reference to our femur-based landmark. This demonstrates an ideal femoro-tibial tracking resulting from our alignment protocol. 

Apart from the appropriate indications to perform a mUKA, it has already been reported in the literature that component malposition in mUKA is the most important risk factor for implant failure [17,18,19,20]. However, the majority of those reports are only related to the potential influence of coronal and sagittal components’ mispositioning and early or late failures [4,5]. Recently, some authors underlined the relationship between components’ rotation alignment and clinical outcomes, showing that a femoro-tibial mismatch can reduce the outcomes [21,22,23]. From the biomechanical perspective a maltracking could also lead to increased wear, edge loading or impingement and in case of mobile bearing UKA also to inlay luxation.

In any case, since the mUKA is a technically demanding procedure and any type of component malposition in any spatial plane can be detrimental to the final outcomes, it is difficult to understand which of these parameters most influences the clinical outcomes and the rates of failures. Therefore, isolating the potential impact of the tibial rotation component mispositioning on the outcomes does not appear to be trivial. For such reasons, we believe that the potential effects of an alteration of the tibial rotation component can be evaluated by considering the dynamic femoro-tibial kinematics of the medial compartment throughout the range of motion. The tracking profile of the tibia on the femur can give us an idea of whether that tibial component rotation is able to recreate an appropriate kinematics, estimating the potential consequences that an impaired femoro-tibial tracking profile can produce from a clinical point of view.

Until the advent of modern navigation systems to perform mUKA, it has never been possible to dynamically record the femoro-tibial tracking profile of the medial compartment intra-operatively. Additional to this, the image-based robotic systems combine these dynamic parameters with the structural data of the CT scan and the patients individual anatomy. Therefore, without the use of robotic-assisted surgery, we can only evaluate retrospectively the consequences that a tibial-rotation mispositioning related to the well-described anatomical references (Akagi line, tibial posterior cortical rim) may have, without knowing if that tibial rotation matched the femur rotation, and thus uncertain of whether we can re-create the good kinematics of the medial compartment of the knee [24,25,26].

With the use of conventional techniques and manual instrumentations, there are several anatomical landmarks that can be used by surgeons to reduce tibial component rotation malposition in mUKA, but none of them are considered a gold-standard [12,27,28,29]. It could be shown, that these landmarks are variable in themselves as well as the correct determination intraoperatively by the attending surgeon is error prone.

The most popular anatomical landmarks for tibial component axial positioning are the tip of the medial tibial spine, the medial sixth of the patellar tendon/medial edge of the tibial tubercle, the medial wall of the intercondylar notch (lateral wall of the medial femoral condyle), and the axis of the anterior superior iliac spine (ASIS). The fact is that some papers show contradictory results from others. Due to the difficulty in identifying the ASIS, Lee et al. [30] did not recommend this anatomical landmark for the guidance of sagittal tibial resection during mUKA, while Kamenaga et al. [23] found the ASIS useful to place the tibial component externally rotated, relative to Akagi’s line that has been correlated with good outcomes. At the same time, Kawahara et al. [12] suggested that the medial sixth of the patellar tendon is an appropriate landmark for the tibial component rotation, but rotating the saw blade during the sagittal resection may produce variable rotational alignment, as dictated by Makhdom et al. [28].

Furthermore, what has emerged from those articles [12,23,28] is not simply that the tibial component malposition could influence the outcome, but that there is substantial variability in tibial component axial positioning. Iriberri et al. [21], in a CT scan-based study, reported a variability of 33° in the axial positioning of the tibial component, much higher than in all the other planes due to the free-hand technique. Even Servien et al. [24] found a range of about 20° (SD ± 10.3) of tibial rotation positioning in only 19 mUKA. In our series of 210 robotic-assisted mUKA, the mean tibial rotation angle was +0.2° (SD ± 2.9°). This great difference may have come from the fact that in conventional mUKA, the rotational alignment of the tibial component is not instrumented, and it is mainly based on variable tibial anatomical landmarks [25]. Positioning the tibial and femoral components independently from each other could introduce a potential rotational mismatch that might lead to suboptimal medial compartment knee kinematics as altered components’ contact points compromises components’ durability [22].

In our study, the tibial component position was set by using a CT scan image-based robotic system that allows the surgeon not only the possibility to pre-operatively match the tibial component with the tibial anatomy, but also to position the tibial component in relation to the position of the femur component. Moreover, this system can evaluate the dynamical femoro-tibial tracking profile throughout the whole ROM, resulting in immediate feedback on how the variation of the tibial rotation can interfere with the kinematics of the medial compartment. We introduced the concept of positioning the tibial component based on the same CT-scan femoral anatomical landmarks (posterior femoral condyle line) we used for the femoral component, allowing the coupling of the two prosthesis components. In the present series, this method to set the tibial rotation was found to be predictable for good medial knee kinematics. The mean FTTA was +0.7° (SD ± 2.2°) with no statistical difference between this angle and the TRA. We believe that coupling the rotation of the tibial component with the femur can be performed in the medial compartment of the knee because it is less affected by the screw-home mechanism [31] than the lateral compartment, and the femoro-tibial tracking profile is almost homogeneous throughout the ROM.

Several authors have already demonstrated the better accuracy of robotic-assisted surgery in component positioning than conventional mUKA, but they mainly focused on the accuracy in coronal and sagittal planes with no reports about the axial plane [10,32,33,34,35]. A recent study by Favroul et al. [36] focused more on the axial positioning of the tibial component in both medial and lateral robotic-assisted UKA. Those authors confirmed our findings and described good kinematics of the medial compartment of the knee setting the tibial rotation by first considering the position of the femur. The difference between their technique and our technique was that they co-aligned the lateral edge of the tibial implant with the axis of the lateral cortex of the medial femoral condyle in mid-flexion, while we co-aligned the tibial component rotation to the posterior femur condyles centered in the middle of the notch preoperatively in the menu to set tibial rotation.

The present study comes with several limitations. First, it is a retrospective series with no control group to evaluate whether other femoral landmarks could be as reliable as our landmarks in reproducing good knee kinematics. Second, we did not use a post-operative CT scan to calculate the rotation of the tibial component concerning the well-known tibial reference lines, such as the Akagi’s line and the tibial posterior cortical rim [24,25,26]. Finally, we based the study only on intra-operative radiological and graphical parameters with no focus on possible clinical consequences of this tibial rotation positioning method. As mentioned above, not only can the tibial rotation lead to poor clinical results, but it can also lead to tibial-component mispositioning in the coronal or sagittal plane, or lead to altered intercompartmental pressures in flexion and extension; therefore, it is difficult to evaluate which component position in which spatial plane influences the clinical outcomes the most. Nevertheless, the presented cases have been followed prospectively for further clinical analyses. Lastly, the results of our study are only verified for the MAKO technology and one specific implant system with the possibility to set the landmarks for tibia rotation based on a preoperative CT. Thus, it might not be applicable for other image-based robotic systems, or other implants, nor is it transferable to image-less technologies. Further studies are necessary to overcome said limitations and to evaluate the clinical effect of tibia rotation in more detail.

Apart from these limitations, this is the first study underlining the importance of setting the tibial rotation concerning femoral landmarks in restoring an optimal medial femoro-tibial tracking profile during the whole ROM. The concept of femur based landmarks can also be interesting for new alignment philosophies in total knee arthroplasty aiming for more natural knee kinematics.

## 5. Conclusions

We found that coupling the tibial and femoral prosthesis component rotations by first setting the tibial rotation based on CT-scan femoral landmarks and not on tibial anatomical landmarks is a reliable method to obtain congruent knee kinematics during robotic-assisted medial UKA. This new method to set the tibial component rotation in mUKA has been shown to be effective in reducing outliers in the combined femoro-tibial component positioning as it avoids any femoro-tibial rotational component mismatch that can potentially alter clinical outcomes.

## Figures and Tables

**Figure 1 jpm-13-00632-f001:**
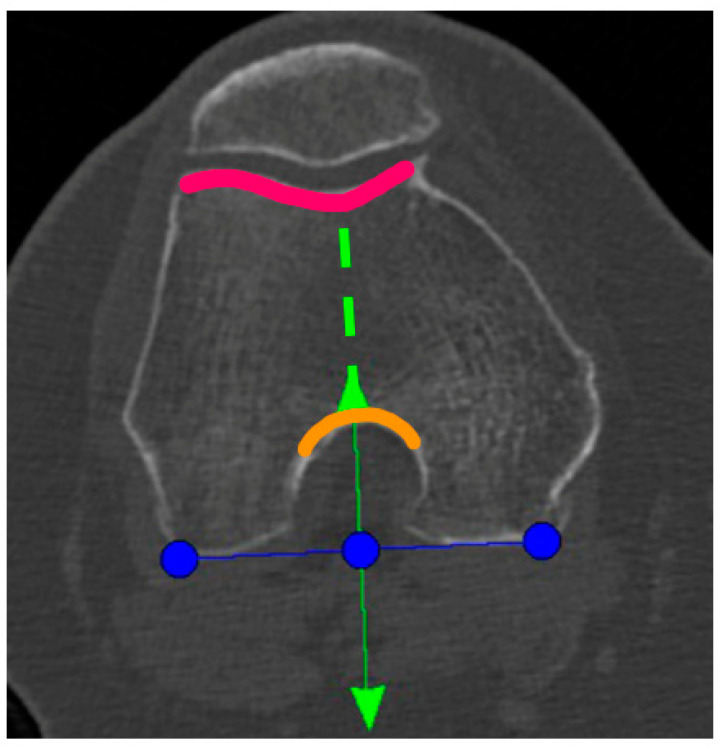
Blue line: tibial rotation landmark co-aligned to the posterior femoral condylar line. Green line: perpendicular to the blue line and centered to the middle of the notch (anterior green arrow and orange curved line) and the trochlea groove (green dotted line and pink curved line).

**Figure 2 jpm-13-00632-f002:**
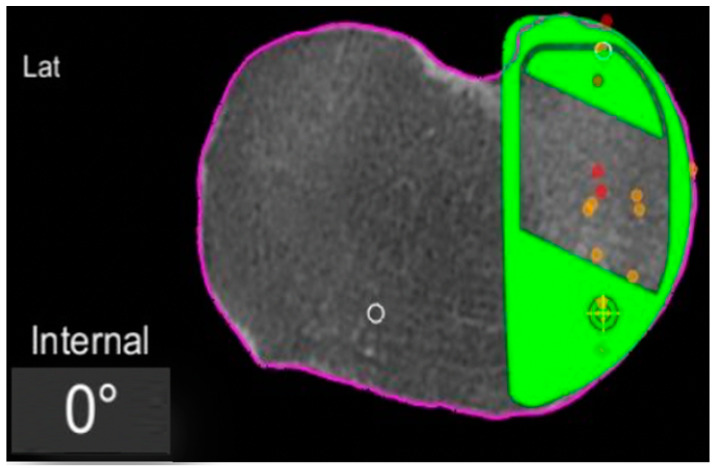
Tibial component rotation (tibial rotation angle, TRA) is set to 0° considering the femur-based rotation landmark. The green area represents the implant. The purple line represents the segmentation of the CT-scan. Red dots are the recorded contact points of the femur on the tibia, yellow dots the cartilage level that was recorded with a blunt probe during surgery, in order to fine-tune the resection level.

**Figure 3 jpm-13-00632-f003:**
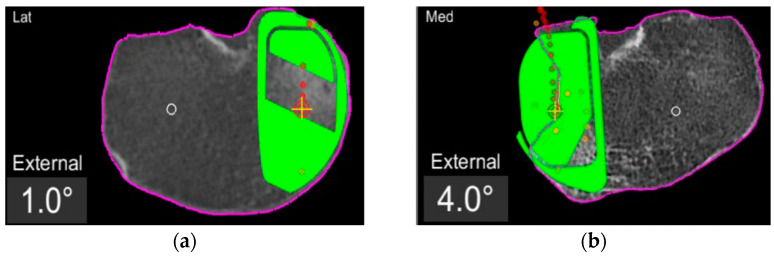
TRA is 1° (**a**) and 4° (**b**) externally rotated to the femur-based rotation landmark.

**Figure 4 jpm-13-00632-f004:**
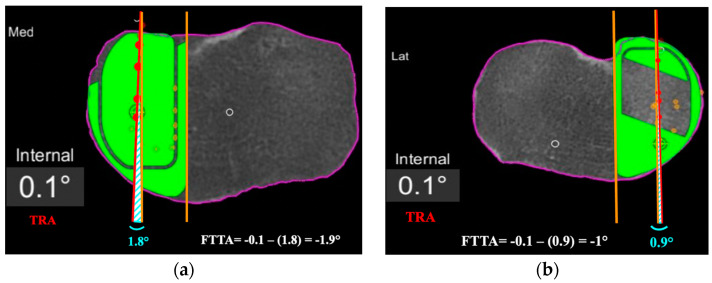
Femoro-tibial tracking angle is calculated by subtracting the angle between a line tangent to the lateral border of the tibial component and a line passing across at least 4 consecutive femoro-tibial tracking points (light blue angle) with the tibial tracking angle (TRA). FTTA is –1.9° in (**a**), and –1° in (**b**).

**Table 1 jpm-13-00632-t001:** FTTA values.

FTTA	n. Cases	Average	Median	SD
//	210	+0.7°	+0.2°	±2.2°
<+1° and >−1°	119	+0°	+0°	±0.1°
<−1°	25	−2.5°	−2.2°	±1.4°
>+1°	66	+3.1°	+2.5°	±2.1°

## Data Availability

The data presented in this study are possibly available on request from the corresponding author. The data are not publicly available due to privacy issues of the participance.

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
