# Peer review of "Setting the Tibial Component Rotation Based on Femoral Landmarks Allows Congruent Knee Kinematics in Robotic-Assisted Medial Unicompartmental Knee Replacement"

_jpm, 2023, doi:10.3390/jpm13040632_

Round 1

Reviewer 1 Report

The authors investigated the effect of image-based robotic-assisted UKA on tibial component rotation. The rotation of the tibial component was based on femoral anatomy rather than bony tibial landmarks to allow for the best possible contact point between femur and tibia. 

The manuscript is written in a logical order. The introduction clearly explains the issue and leads nicely to the research question. This study's hypothesis has been developed.

The methodology is consistent and well-explained.

The discussion connects the investigated relationships to current literature and critically examines them. The limitations are stated.

Overall, this study investigates an intriguing correlation that provides a basis for additional studies in which non-"robotic surgeons" may benefit as well. As a result, the findings of this study will have an impact on the clinical activities of UKA surgeons.

Author Response

thanks for the feedback - no changes required.

Reviewer 2 Report

This is a very good and sophisticated study to a high matter of concern. One limitation may be, that your (good) results are not transferable to other robotic assisted systems other than Mako. Please state that comment in your limitation section.

Author Response

thanks for the feedback - the limitation was added to the manuscript as suggested

Reviewer 3 Report

1.      The Reviewer do not see the novel in the present article. My examination revealed that several similar previous publications appear to appropriately address the issues you have brought up in the current submission. Please emphasize it more advance in the introduction section if there are any more truly something really new.

2.      Previous study related needs to explain in the introduction section consisting of their work, their novelty, and their limitations to show the research gaps that intend to be filled in the present study.

3.      Line 200, the authors mentioned using CT scan image based. Please refer and discuss the concept and procedure in more detail. Similar research conducted with CS scan image needs to adopted for this purpose as follows: The Effect of Tortuosity on Permeability of Porous Scaffold. Biomedicines 2023, 11, 427. https://doi.org/10.3390/biomedicines11020427

4.      Line 216-234 would be better to combine into one paragraph.

5.      Line 236-238, please develop the conclusion section.

6.      The conclusion section needs to explain further research.

7.      In the whole of the manuscript, the authors sometimes made a paragraph only consisting of one or two sentences that made the explanation not clearly understood. The authors need to extend their explanation to become a more comprehensive paragraph. In one paragraph, it is recommended to consist of at least 3 sentences with 1 sentence as the main sentence and the other sentences as supporting sentences.

Author Response

Sorry, but this review seem not to belong to our manuscript??
The noted reference is completely off the topic and also the other comments

- no changes

Round 2

Reviewer 3 Report

The previous comment is for the author's manuscript. Please refer it point by point along with an explanation and changes made. Without addressing all of them, the present manuscript is inappropriate for publication and should be rejected. Thanks.